# Gut Microbiota Alleviates Intestinal Injury Induced by Extended Exposure to Light via Inhibiting the Activation of NLRP3 Inflammasome in Broiler Chickens

**DOI:** 10.3390/ijms25126695

**Published:** 2024-06-18

**Authors:** Dandan Ma, Minhong Zhang, Jinghai Feng

**Affiliations:** State Key Laboratory of Animal Nutrition and Feeding, Institute of Animal Sciences, Chinese Academy of Agricultural Sciences, Beijing 100193, China; madandan93@163.com (D.M.); fengjinghai@caas.cn (J.F.)

**Keywords:** extended exposure to light, gut microbiota, intestinal inflammation, NLRP3 inflammasome, broiler chickens

## Abstract

Light pollution is a potential risk for intestinal health in humans and animals. The gut microbiota is associated with the development of intestinal inflammation induced by extended exposure to light, but the underlying mechanism is not yet clear. The results of this study showed that extended exposure to light (18L:6D) damaged intestinal morphology, downregulated the expression of tight junction proteins, and upregulated the expression of the NLRP3 inflammasome and the concentration of pro-inflammatory cytokines. In addition, extended exposure to light significantly decreased the abundance of *Lactobacillus, Butyricicoccus*, and *Sellimonas* and increased the abundance of *Bifidobacterium*, *unclassified Oscillospirales*, *Family_XIII_UCG-001*, *norank_f__norank_o__Clostridia_vadinBB60_group*, and *Defluviitaleaceae_UCG-01*. Spearman correlation analysis indicated that gut microbiota dysbiosis positively correlated with the activation of the NLRP3 inflammasome. The above results indicated that extended exposure to light induced intestinal injury by NLRP3 inflammasome activation and gut microbiota dysbiosis. Antibiotic depletion intestinal microbiota treatment and cecal microbiota transplantation (CMT) from the 12L:12D group to 18L:6D group indicated that the gut microbiota alleviated intestinal inflammatory injury induced by extended exposure to light via inhibiting the activation of the NLRP3 inflammasome. In conclusion, our findings indicated that the gut microbiota can alleviate intestinal inflammation induced by extended exposure to light via inhibiting the activation of the NLRP3 inflammasome.

## 1. Introduction

Currently, global nighttime lighting is increasing, especially in urban environments and even in remote areas [1]. A recent study showed that over the past 12 years, the range of nighttime electricity usage has increased by nearly 10% annually, far exceeding previous expectations [2]. This means that the daily exposure of humans and animals to light has been extended. An increasing number of studies have pointed out the adverse effects of prolonged exposure to sunlight on human and animal health, including the risk of chronic diseases, but this knowledge is still in its early stages [1].

Abnormal lighting exposure (such as light at night and excessive light extensity) not only leads to sleep disorders [3], emotional disorders, and severe depression [4], but also affects intestinal health [5,6]. Recently, an increasing number of studies have shown that prolonged exposure to light leads to intestinal dysfunction. An increased level of TNF-α was found in layer ducks under the condition of prolonged exposure to light [7]. In addition, a previous study in our laboratory found that extended exposure to light (23L:1D) significantly increased the content of inflammatory cytokines in the intestine and disrupted the morphology of intestine tissue, which leads to intestinal inflammation in broiler chickens [8].

More and more evidence suggests that the imbalance of the gut microbiota is related to the development of intestinal inflammation [9,10,11]. Prolonging exposure to light can alter the composition of the gut microbiota in animals [12,13,14,15]. Interestingly, previous research found that the chicks in a 12L:12D group possessed genera that are typically associated with healthy guts, whereas the chicks in a 23L:1D group possessed genera that are typically found in diseased guts [16]. It was found that extended exposure to light could decrease beneficial bacteria and increase harmful bacteria, and then induce intestinal inflammation in broiler chickens [15,16], which indicates the important role of the gut microbiota under the condition of extended exposure to light. However, the mechanism of the influence of the gut microbiota on intestinal inflammation induced by extended exposure to light is still unclear.

NOD-like receptor thermal protein domain-associated protein 3 (NLRP3) is a crucial pattern recognition receptor (PRR), which, together with adaptor protein ASC and procaspase-1 precursor, forms the NLRP3 inflammasome [17,18]. NLRP3 is the most representative inflammasome [19]. When the NLRP3 inflammasome is activated by intracellular and extracellular stimuli, procaspase-1 is activated to caspase1, which activates the inflammatory cytokines (IL-1β and IL-18), ultimately leading to an inflammatory response [20]. In recent years, the role of the NLRP3 inflammasome in intestinal diseases has attracted widespread attention. NLRP3 inflammasomes and their downstream signaling molecules have become potential targets for the treatment of various intestinal diseases in chickens [20,21]. Therefore, the NLRP3 inflammasome plays an important role in intestinal health.

The gut microbiota can recognize immune regulatory proteins such as anti-inflammatory cytokines and pro-inflammatory cytokines through PRRs, which play important roles in inflammatory responses [22]. In a previous study in our laboratory, it was found that under the condition of extended exposure to light, the NLRP3 inflammasome may interact with the gut microbiota, leading to intestinal inflammation in broiler chickens [8]. Thus, we hypothesized that the NLRP3 inflammasome may be an important pathway between the gut microbiota and intestinal inflammation induced by extended exposure to light.

Gut microbiota dysbiosis is an important factor leading to intestinal inflammation [23]. The process of transforming the gut microbiota through microbiota transplantation, which involves transferring the gut microbiota from healthy donors to patients for the treatment of microbial dysbiosis, is receiving much attention from the medical community [24,25]. Thus, we employed cecal microbiota transplantation (CMT) and antibiotic treatment to validate the hypothesis that the gut microbiota alleviates the intestinal inflammation induced by extended exposure to light in a model of broiler chickens, which provides new insights into the impact of light pollution on intestinal health.

## 2. Results

### 2.1. Extended Exposure to Light Caused Intestinal Injury and Impaired Intestinal Barrier

To determine the effect of extended exposure to light on the tissue structure of the intestine, HE staining was used to examine the duodenum morphology. Compared with the 12L:12D group, the intestine inflammation was significant: epithelial cell shedding and inflammatory cells increased in the 18L:6D group (Figure 1A,B). In addition, extended exposure to light significantly decreased the mRNA expression of *Claudin-1*, *Occludin*, and *ZO-1* (Figure 1C–E). Thus, extended exposure to light caused intestinal injury and an impaired intestinal barrier.

### 2.2. Extended Exposure to Light Activated NLRP3 Inflammasome and Increased Pro-Inflammatory Cytokines

Extended exposure to light significantly increased the mRNA expression of *NLRP3*, *caspase1*, and *IL-1β* (Figure 2A–C). In addition, extended exposure to light significantly increased the concentration of pro-inflammatory cytokines (IL-1β, IL-6, IL-18, and TNF-α) (Figure 2D–G) in the duodenum of the broiler chickens. Thus, extended exposure to light activated the NLRP3 inflammasome and increased pro-inflammatory cytokines.

### 2.3. Extended Exposure to Light Altered the Gut Microbiota Composition

As indicated in Figure 3A, principal coordinate analysis (PCoA) of OTUs related to β-diversity revealed significant differences in the cecal content of the microbiota between the 12L:12D group and 18L:6D group. Firmicutes, Bacteroidetes, Proteobacteria, and Actinobacteria constituted the majority of the gut microbiota at the phylum level (Figure 3B). In addition, significant changes in the gut microbiota composition were detected at the genus level (Figure 3C). The differences in gut microbiota composition between the 12L:12D group and 18L:6D group (Figure 3D) showed that the relative abundance of *Lactobacillus*, *Butyricicoccus*, and *Sellimonas* was significantly decreased, and the relative abundance of *Bifidobacterium*, *Clostridia_vadin group*, *unclassified Defluviitaleaceae*, and *unclassified Oscillospirales* was significantly increased in the 18L:6D group (*p* < 0.05).

### 2.4. Correlation Analysis between the Gut Microbiota and NLRP3 Inflammasome in 12L:12D Group and 18L:6D Group

The Spearman correlation coefficient was used to analyze the correlation between the gut microbiota and NLRP3 inflammasome (Figure 4). The relative abundance of *Lactobacillus* was negatively correlated with the mRNA expression level of *IL-1β* (*p* < 0.05). The relative abundance of *Sellimonas* was negatively correlated with the mRNA expression level of *NLRP3* (*p* < 0.05). The relative abundance of *Bifidobacterium* and *unclassified Oscillospirales* was positively correlated with the mRNA expression level of *NLRP3* (*p* < 0.05). The relative abundance of *Family_XIII_UCG-001* was positively correlated with the mRNA expression level of *caspase1* and *IL-1β* (*p* < 0.05). The relative abundance of *norank_f__norank_o__Clostridia_vadinBB60_group* was positively correlated with the mRNA expression level of *caspase1* (*p* < 0.05). The relative abundance of *unclassified Defluviitaleaceae* was positively correlated with the mRNA expression level of *NLRP3* and *caspase1* (*p* < 0.05).

### 2.5. Cecal Microbiota Transplantation Alleviated Intestinal Injury and Inhibited the Activation of NLRP3 Inflammasome

As shown in Figure 5A–C, compared with the 12L:12D group, inflammatory injury in the duodenum was found in the 18L:6D group and 18L:6D+PBS group. Compared with the 18L:6D group and 18L:6D+PBS group, cecal gut microbiota transplantation alleviated intestinal injury and inflammatory infiltration (Figure 5B–D). For tight junction proteins, cecal microbiota transplantation significantly reversed the effect of extended exposure to light on the changes in the mRNA expression of *Occludin* and *ZO-1*. In addition, compared with the 12L:12D group, the mRNA expression levels of *NLRP3*, *caspase1*, and *IL-1β* were significantly increased in the 18L:6D group and 18L:6D+PBS group (*p* < 0.05). The mRNA expression levels of *NLRP3*, *caspase1*, and *IL-1β* were significantly decreased in the 18L:6D+CMT group compared with the 18L:6D group and 18L:6D+PBS group (*p* < 0.05) (Figure 5E–G). Cecal microbiota transplantation significantly reversed the effect of extended exposure to light on the changes in the concentration of pro-inflammatory cytokines (IL-1β, IL-6, IL-18, and TNF-α) (*p* < 0.05). These results indicated that the cecal microbiota plays an important role in the mitigation of intestinal inflammation induced by extended exposure to light via inhibiting the activation of the NLRP3 inflammasome.

### 2.6. Antibiotic Treatment Alleviated Intestinal Injury and Inhibited the Activation of NLRP3 Inflammasome

As shown in Figure 6, compared with the 18L:6D group, the intestinal tissue was normal, and no obvious infiltration of inflammatory cells was observed in the antibiotic treatment group (Figure 6B,C). Regarding tight junction proteins, the antibiotic treatment significantly reversed the effect of extended exposure to light on the changes in the mRNA expression of *Occludin* (*p* < 0.05) (Figure 6E). In addition, the mRNA expression level of the NLRP3 inflammasome (*NLRP3*, *caspase1*, and *IL-1β*) (Figure 6G–I) and the concentration of pro-inflammatory cytokines (IL-1β, IL-6, IL-18, and TNF-α) were significantly decreased in the antibiotic treatment group (*p* < 0.05) (Figure 6J–M). These results indicated that the antibiotic treatment alleviated the intestinal inflammation and inhibited the activation of the NLRP3 inflammasome induced by extended exposure to light.

## 3. Discussion

Light pollution is a potential risk factor for animal and human health [26]. It has been reported that prolonged exposure to light can lead to intestinal inflammation [8,15]. However, the understanding of its underlying mechanism is not comprehensive. Here, our study first demonstrated the role of the gut microbiota in alleviating intestinal inflammatory injury induced by extended exposure to light through the NLRP3 inflammasome.

Extended exposure to light may also have a direct or indirect impact on the gut microbiota, which may also lead to intestinal inflammation.

More and more evidence suggests that gut microbiota dysbiosis plays an important role in intestinal inflammation [27,28]. In this study, we found that extended exposure to light significantly decreased the relative abundance of *Lactobacillus*, *Butyricicoccus*, and *Sellimonas* and increased the relative abundance of *Bifidobacterium*, *Clostridia_vadin group*, *unclassified Defluviitaleaceae*, and *unclassified Oscillospirales*. *Lactobacillus*, one of the most abundant bacteria in the gut microbiota, was shown to be related to intestinal health [29]. Research has found a significant decrease in the abundance of *Lactobacillus* in patients with irritable bowel syndrome (IBS) [30]. *Butyricicoccus* can produce butyrate, which plays an important role in maintaining intestinal barrier function and has anti-inflammatory properties [31,32,33]. The abundance of *Sellimonas* was decreased in dogs with inflammatory bowel disease [34]. When inflammation occurs, the abundance of *Family_XIII_UCG-001* was increased [35]. The results of this study indicated that extended exposure to light decreased the abundance of beneficial bacteria and increased the abundance of harmful bacteria, leading to gut microbiota dysbiosis. In addition, extended exposure to light damaged intestinal morphology. Therefore, our study suggested that gut microbiota dysbiosis associated with extended exposure to light leads to intestinal inflammation.

The NLRP3 inflammasome plays an important role in maintaining the stability of the intestinal immune system [36,37]. When the NLRP3 inflammasome is abnormally activated, various intestinal diseases can be induced [38,39,40,41]. NLRP3 expression was increased in the intestine tissue of a model of intestinal inflammation in chickens [20,42,43]. In our study, we found that extended exposure to light induced intestinal injury. In addition, extended exposure to light activated the NLRP3 inflammasome, suggesting that extended exposure to light induced intestinal inflammation. In addition, we used the Spearman correlation coefficient to analyze the correlation between the gut microbiota and NLRP3 inflammasome and found a possible interaction between the gut microbiota and NLRP3 inflammasome. These results indicated that intestinal inflammation induced by extended exposure to light is related to the gut microbiota and activation of the NLRP3 inflammasome.

Restoring immune homeostasis through the normalization of the gut microbiota is now considered a valuable treatment method for treating intestinal inflammation [27]. Microbial targeted therapy, including antibiotics, probiotics, and microbiota transplantation, is considered an effective treatment for intestinal inflammation. In our study, we demonstrated the relationship between the gut microbiota and intestinal inflammation using antibiotic treatment and cecal microbiota transplantation. Clearing the gut microbiota of mice with broad-spectrum antibiotics alleviated neuroinflammation and inhibited the activation of the NLRP3 inflammasome induced by chronic ethanol exposure [44], which was consistent with the inhibition of the NLRP3 inflammasome after antibiotic treatment in the intestine. It was reported that gut microbiota dysbiosis activated the NLRP3 inflammasome in the colon and brain, mediating the inflammatory response induced by chronic unpredictable mild stress [45]. In mice exposed to chronic ethanol, it was found that gut microbiota dysbiosis leads to neuroinflammation by activating the NLRP3 inflammasome in the hippocampus [46]. The above three studies indicate that the gut microbiota in mammals can regulate NLRP3 inflammasomes, thereby affecting the inflammatory process. However, this mechanism has not yet been found in birds. Thus, we used cecal microbiota transplantation and antibiotic treatment to validate the function of the gut microbiota in alleviating intestinal inflammation via the NLRP3 inflammasome under the condition of extended exposure to light in broiler chickens. We found that the gut microbiota alleviated intestinal inflammation induced by extended exposure to light via inhibiting NLRP3 inflammasome activation in broiler chickens.

## 4. Materials and Methods

### 4.1. Photoperiod Model

All animal experiments were approved by the Animal Ethics Committees of the Institute of Animal Sciences, Chinese Academy of Agricultural Sciences (Ethics Code Permit: IAS2021-245). One-day-old male broiler chickens were obtained from a local hatchery and raised in an environmentally controlled room for 4 days to adapt to the environment chamber before the photoperiod experiment. At the age of 5 days, 96 male broiler chickens were divided into two groups (12L:12D group and 18L:6D group), with 6 replicates per group and 8 broiler chickens per replicate. The setting of the photoperiod is shown in Table 1. The broiler chickens were reared in environmentally controlled chambers of the State Key Laboratory of Animal Science and given ad libitum access to feed and water for 3 weeks. During the whole experiment, the basal diet was formulated to meet the National Research Council’s (NRC, 1994) recommended requirement.

At the end of the experiment, one male broiler chicken from every replicate of the 12L:12D group and 18L:6D group was euthanized by cervical dislocation, and duodenum and cecal contents were collected. Duodenum tissues were used to analyze histological changes and detect the mRNA expression of *NLRP3*, *caspase1*, and *IL-1β*. Cecal contents were used to analyze the changes in microbiota composition.

### 4.2. Antibiotic Treatment and Cecal Microbiota Transplantation (CMT)

To determine whether the gut microbiota plays an important role in the intestinal inflammation caused by extended exposure to light, antibiotic treatment and cecal microbiota transplantation were conducted. One-day-old male broiler chickens were obtained from a local hatchery and raised in an environmentally controlled room for 4 days to adapt to the environment chamber before the antibiotic treatment and cecal microbiota transplantation experiments. At the age of 5 days, a total of 100 male broiler chickens were randomly and evenly divided into five groups: the 12L:12D, 18L:6D, 18L:6D+Antibiotics, 18L:6D+PBS (negative control group), and 18L:6D+CMT (cecal microbiota transplantation) group. The setting of the photoperiod was the same as that of the photoperiod model experiment. The broiler chickens in the 12L:12D group and 18L:6D group were fed with normal diet for 3 weeks. In the 18L:6D+Antibiotics group, the broiler chickens were fed with combinatorial antibiotics (100 mg/g streptomycin, 100 mg/g vancomycin, 100 mg/g metronidazole, 100 mg/g amoxicillin) for 3 weeks to deplete the gut microbiota. Our laboratory has demonstrated the effect of the above combinational antibiotics on depleting the gut microbiota [28]. In the 18L:6D+CMT group, the gut microbiota of the broiler chickens from the 12L:12D group was transplanted to the broiler chickens of the 18L:6D group. The cecal contents were collected from the 12L:12D group of the photoperiod model experiment, suspended in PBS, and centrifuged. Then, the supernatant was collected. The broiler chickens of the CMT group received the supernatant of cecal contents, and the broiler chickens of the 18L:6D+PBS group received PBS for 10 days. All broiler chickens were reared in environmentally controlled chambers of the State Key Laboratory of Animal Science and given ad libitum access to feed and water for 3 weeks. During the whole experiment, the basal diet was formulated to meet the National Research Council’s (NRC, 1994) recommended requirement.

At the end of the experiment, six male broiler chickens from every group were euthanized by cervical dislocation, and duodenum tissue samples were collected to detect histological changes, pro-inflammatory cytokines (IL-1β, IL-6, IL-18, and TNF-α), and mRNA expression of the NLRP3 inflammasome (NLRP3, caspase1, and IL-1β) and tight junction proteins (Claudin-1, Occludin, and ZO-1).

### 4.3. Histochemical Analysis

Duodenum tissues were fixed overnight in 4% polyformaldehyde and processed for embedding. Then, the cross sections were cut into 6 μm slices, and the slices were placed onto glass slides. Then, the slices were dewaxed in xylene and washed with gradient ethanol before hydration. Hematoxylin and eosin staining was performed according to standard procedures.

### 4.4. Inflammatory Cytokines

The concentration levels of IL-1β (H002), IL-6 (H007), IL-18 (H015), and TNF-α (H052) in serum and duodenum of the broiler chickens (n = 6) were measured using ELISA kits, according to the manufacturer’s instructions (Nanjing Jiancheng Bioengineering Institute, Nanjing, China).

### 4.5. Real-Time Quantitative PCR

Real-time quantitative PCR was used to measure the mRNA expression of the NLRP3 inflammasome (*NLRP3*, *caspase1*, and *IL-1β*) and tight junction proteins (*Claudin-1*, *Occludin*, and *ZO-1*) of the duodenum. Trizol (Invitrogen, Carlsbad, CA, USA) was used to extract total RNA from duodenum tissue of the broiler chickens. The primers of target genes are listed in Table 2.

### 4.6. 16S rRNA Gene Sequence Analysis

According to the manufacturer’s instructions, total microbial genomic DNA was extracted from cecal content samples of the broiler chickens using the E.Z.N.A.^®^ soil DNA Kit (Omega Bio-tek, Norcross, GA, USA). The extracted DNA as template, the V3-V4 region of the 16S rRNA, was amplified with primer pairs 338F(5′-ACTCCTACGGGAGGCAGCAG-3′) and 806R(5′-GGACTACHVGGGTWTCTAAT-3′) [47]. The raw data were uploaded to NCBI Sequence Read Archive Database (Accession Number: PRJNA1019873). A detailed method was described previously [48].

### 4.7. Statistical Analysis

Differences in the mRNA expression of *NLRP3*, *caspase1*, *IL-1β*, *Claudin-1*, *Occludin* and *ZO-1* were analyzed using SPSS Statistics 17.0. Differences with *p* < 0.05 were considered significant.

Bioinformation analysis of the gut microbiota was carried out using the Majorbio Cloud platform (https://cloud.majorbio.com). The principal coordinate analysis was performed based on Bray–Curtis dissimilarity, and the significance in the microbial community was analyzed using the Wilcoxon rank-sum test. The correlation between the gut microbiota and NLRP3 inflammasome was analyzed based on the Spearman correlation.

## 5. Conclusions

In conclusion, our results indicated that extended exposure to light caused intestinal injury, the activation of the NLRP3 inflammasome, and gut microbiota dysbiosis. In addition, antibiotic treatment and cecal gut microbiota transplantation demonstrated that the gut microbiota could alleviate intestinal inflammation induced by extended exposure to light via inhibiting the activation of the NLRP3 inflammasome. Our findings provide a new theoretical basis for exploring the role of the gut microbiota in intestinal inflammatory injury and provide a new regulatory target for alleviating the intestinal inflammation induced by extended exposure to light.

## Figures and Tables

**Figure 1 ijms-25-06695-f001:**
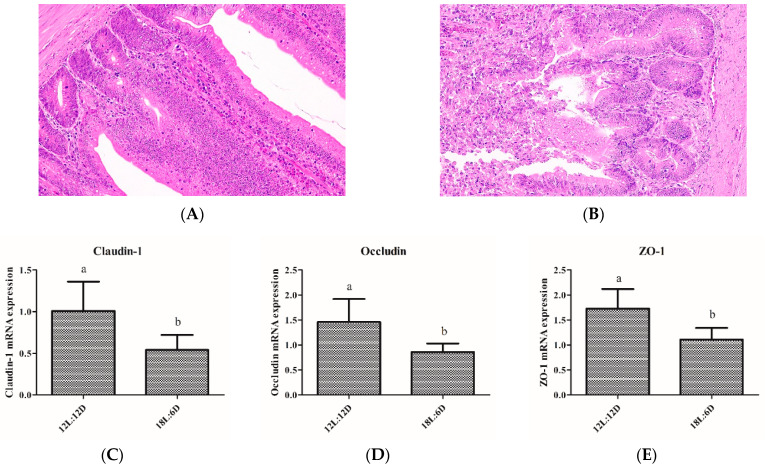
Extended exposure to light caused intestinal injury and decreased the mRNA expression of tight junction protein. (**A**) representative histological images of intestinal tissue of 12L:12D group was captured by a microscope (NIKON DS-U3) at 200×; (**B**) representative histological images of intestinal tissue of 18L:6D group was captured by a microscope (NIKON DS-U3) at 200×. (**C**–**E**) mRNA expression of *Claudin-1*, *Occludin*, and *ZO-1* (n = 6). There were significant differences between different lower-letters (a and b) (*p* < 0.05), but no significant differences between the same lower-letters (*p* > 0.05).

**Figure 2 ijms-25-06695-f002:**
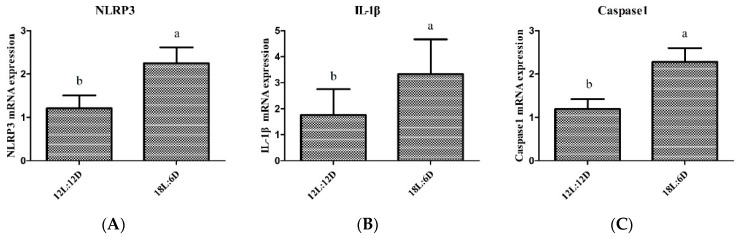
Extended exposure to light activated NLRP3 inflammasome and increased pro-inflammatory cytokines (n = 6). (**A**–**C**) mRNA expression of *NLRP3*, *caspase1* and *IL-1β* in the duodenum of broiler chickens. (**D**–**G**) the concentration of pro-inflammatory cytokines (IL-1β, IL-6, IL-18, and TNF-α) in the duodenum of broiler chickens. There were significant differences between different lower-letters (a and b) (*p* < 0.05), but no significant differences between the same lower-letters (*p* > 0.05).

**Figure 3 ijms-25-06695-f003:**
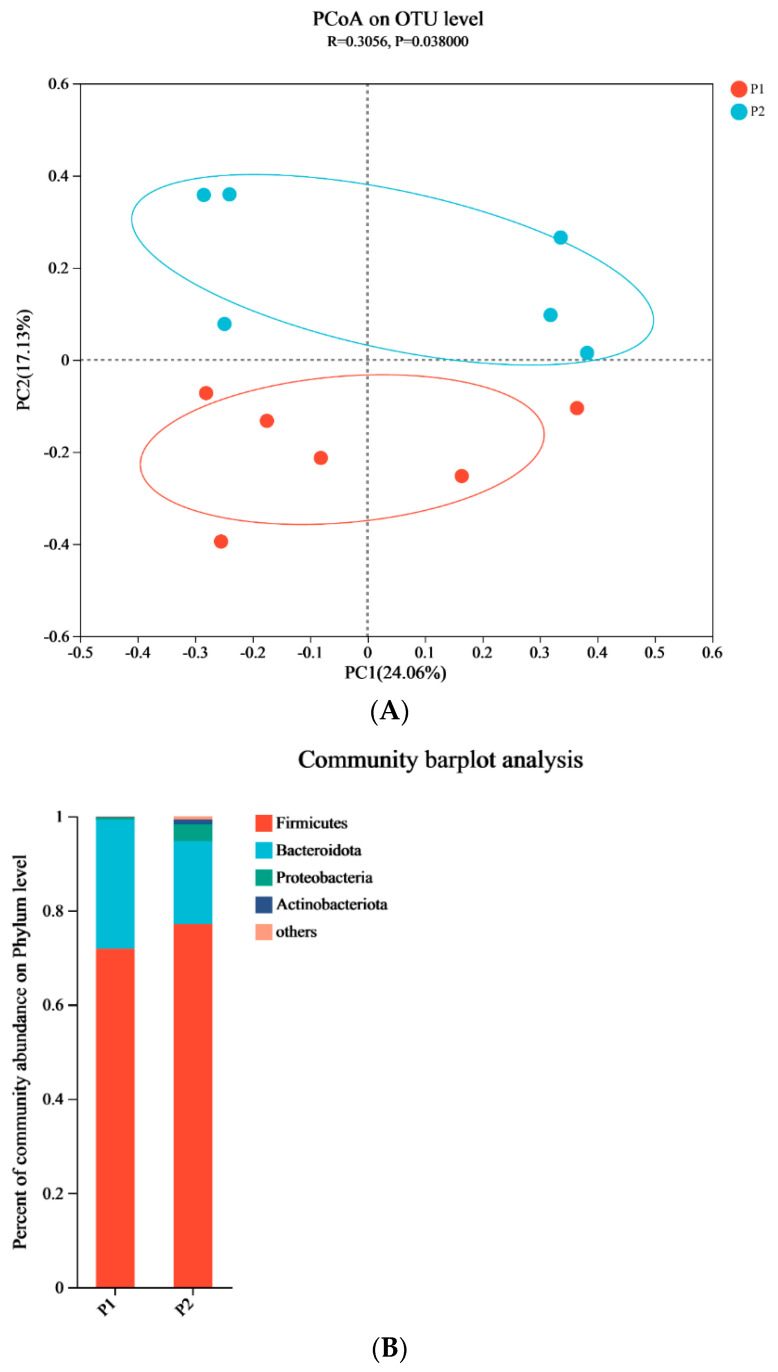
Extended exposure to light altered gut microbiota composition. (**A**) PCoA analysis of gut microbiota; (**B**) the abundance of the gut microbiota at the phylum level; (**C**) the abundance of the gut microbiota at the genus level; (**D**) differences in gut microbiota composition at the genus level between 12L:12D group and 18L:6D group. P1 refers to 12L:12D group; P2 refers to 18L:6D group. * and ** Indicate a significant correlation between two indicators (* indicates *p* < 0.05 and ** indicates *p* < 0.01, n = 6).

**Figure 4 ijms-25-06695-f004:**
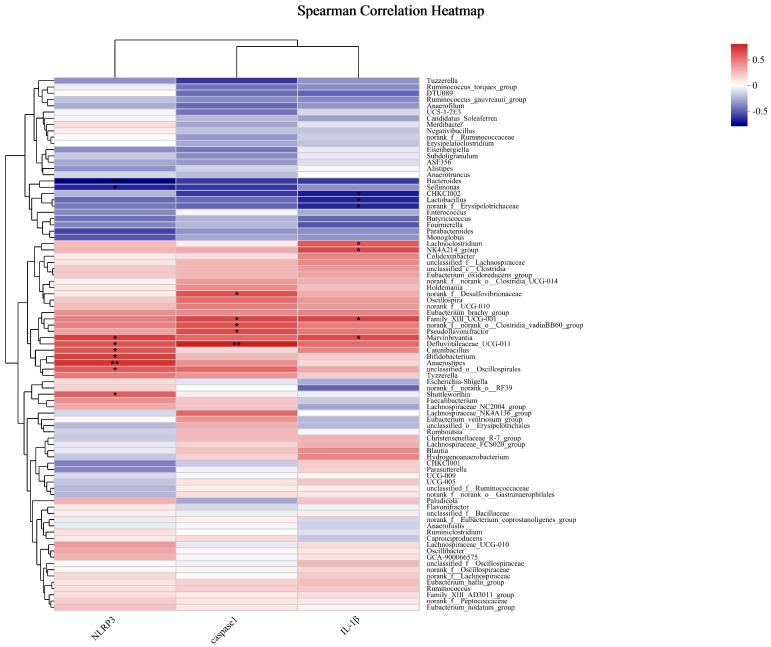
Correlation analysis between gut microbiota and NLRP3 inflammasome at the genus level. Note: the red and blue matrices represent positive and negative correlations, respectively, and the color depth represents the magnitude of the correlation coefficient. * and ** Indicate a significant correlation between two indicators (* indicates *p* < 0.05 and ** indicates *p* < 0.01, n = 6).

**Figure 5 ijms-25-06695-f005:**
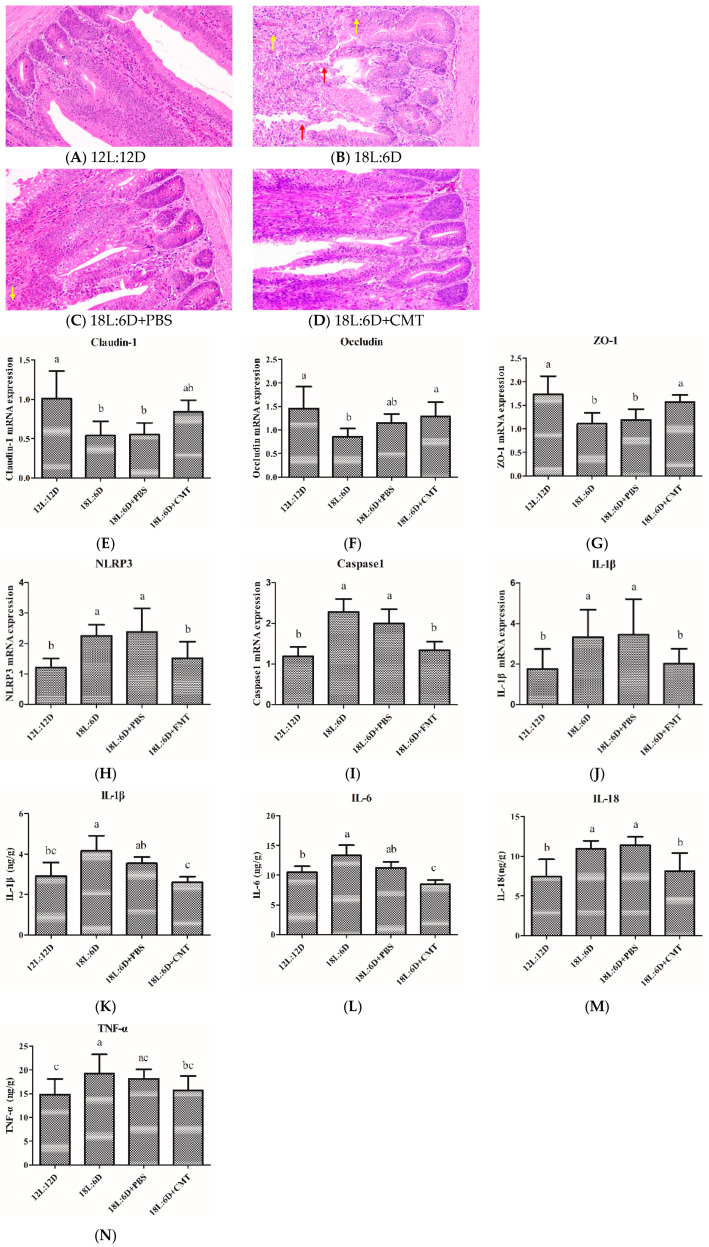
Cecal microbiota collected from broiler chickens of 12L:12D group transplanted to the broiler chickens of 18L:6D group alleviated intestinal inflammation (n = 6). (**A**–**D**) representative histological images of intestinal tissue of 12L:12D, 18L:6D, 18L:6D+PBS, and 18L:6D+CMT groups were captured by a microscope (NIKON DS-U3) at 200×(the red arrows indicate the erosive intestinal villi and the shed epithelial cells of the mucosal layer; the yellow arrows indicate the infiltration of inflammatory cells). (**E**–**G**) mRNA expression of *Claudin-1*, *Occludin*, and *ZO-1*. (**H**–**J**): mRNA expression of *NLRP3*, *caspase1*, and *IL-1β*. (**K**–**N**) the concentration of IL-1β, IL-6, IL-18, and TNF-α. There were significant differences between different lower-letters (a, b and c) (*p* < 0.05), but no significant differences between the same lower-letters (*p* > 0.05).

**Figure 6 ijms-25-06695-f006:**
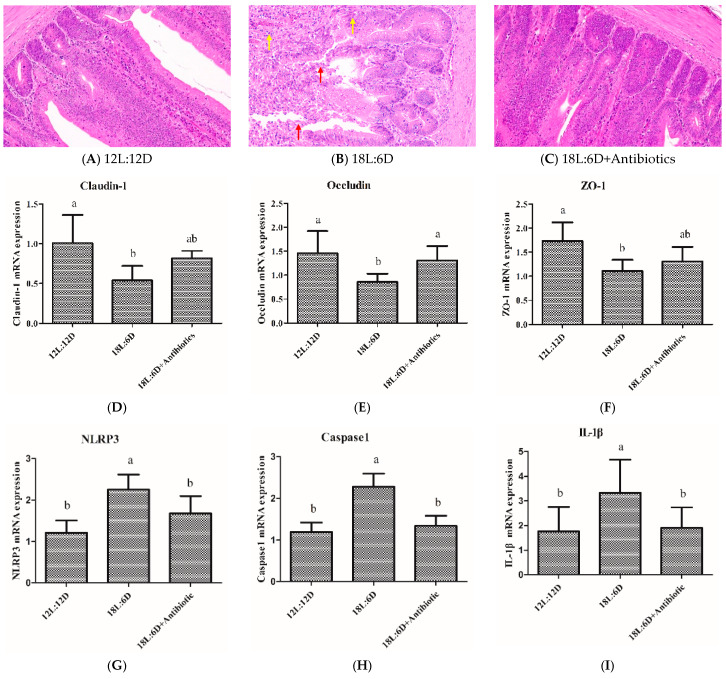
Antibiotic treatment alleviated intestinal injury and inhibited the activation of NLRP3 inflammasome induced by extended exposure to light (n = 6). (**A**–**C**) representative histological images of intestinal tissue of 12L:12D group, 18L:6D group, and 18L:6D+Antibiotic treatment group were captured by a microscope (NIKON DS-U3) at 200× (the red arrows indicate the erosive intestinal villi and the shed epithelial cells of the mucosal layer; the yellow arrows indicate the infiltration of inflammatory cells). (**D**–**F**) mRNA expression of *Claudin-1*, *Occludin*, and *ZO-1*. (**G**–**I**) mRNA expression of *NLRP3*, *caspase1*, and *IL-1β*. (**J**–**M**) the concentration of IL-1β, IL-6, IL-18, and TNF-α. There were significant differences between different lower-letters (a, b and c) (*p* < 0.05), but no significant differences between the same lower-letters (*p* > 0.05).

**Table 1 ijms-25-06695-t001:** Photoperiod setting of 12L:12D group and 18L:6D group.

Day	Experimental Phase	Light Intensity	Photoperiod	
			12L:12D	18L:6D
1–3 d	Adaptive phase	From 50 to 20 lux	24L (Lights on all day)	24L (Lights on all day)
4 d	Transitional phase		18L:6D (Lights on at 00:00; lights off at 18:00)	21L:3D (Lights on at 03:00; lights off at 05:00)
5–26 d	Formal experiment phase	15 lux	12L:12D (Lights on at 06:00; lights off at 18:00)	23L:1D (Lights on at 06:00; lights off at 05:00 the next day)

**Table 2 ijms-25-06695-t002:** The primer sequences of NLRP3, caspase1, and IL-1β for real-time quantitative PCR.

Gene Name	Accession Number	Primer Sequence (5′-3′)	Product Length (bp)
*Actin*	NM_205518.1	Forward: CTGACTGACCGCGTTACTCC Reverse: TTGCACATACCGGAGCCATT	84
*NLRP3*	NM_001348947.2	Forward: CTGAAGTGCCTGGACCTGAGT Reverse: TGTAGAAGTGCTCAGCCCCAG	115
*Caspase1*	XM_015295935.4	Forward: TGCTGCCGTGGAGACAACAT Reverse: CACTGTTAAAGGCATGGTTTCG	235
*IL-1β*	NM_204524.2	Forward: TACATGTCGTGTGTGATGAGCG Reverse: TGGTCGGGTTGGTTGGTGAT	224
*Claudin-1*	NM_001013611.2	Forward: ATGGAGGATGACCAGGTGAAGAA Reverse: TCCAAACTCAAATCTGGTGTTAACG	165
*Occludin*	NM_205128.1	Forward: GAGTTGGATGAGTCCCAGTATGAG Reverse: ATTGAGGCGGTCGTTGATG	204
*ZO-1*	XM_015278981.2	Forward: GGAGGATCCAGCCATGAAAC Reverse: CTTGAGGTCTCTGTGGTTCTGG	236

## Data Availability

The raw data of 16S gene sequence was uploaded to NCBI Sequence Read Archive Database (Accession Number: PRJNA1019873).

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
