# Peer review of "Gut Microbiota Alleviates Intestinal Injury Induced by Extended Exposure to Light via Inhibiting the Activation of NLRP3 Inflammasome in Broiler Chickens"

_ijms, 2024, doi:10.3390/ijms25126695_

Round 1

Reviewer 1 Report

Comments and Suggestions for Authors

The authors present the  results obtained regarding the influence of extended exposure to light on broiler chickens. In my opinion experiments made and the results obtained are exciting, the authors performed many studies with interesting results but in this form, the manuscript cannot be published.
I encourage the authors to read again with attention the manuscript and to solve the following aspects

1) At the introduction: the authors must perform an introduction in connection with their experiments  (they must consider only the studies conducted on birds, not on human patients and not by mice). The introduction must be rewritten.

2)Before the beginning of the chapter entitled ''Materials and Methods'' the authors must write the objectives of the experimental studies presented in the manuscript.

3) At results: Figure 4 is unclear. The authors must upload here a figure with a high resolution.

4) The chapter entitled ''Discussions'' must be completely rewritten. That is because, in the experimental part,  the authors have performed their studies only on chicken broilers  (birds), so the discussions must be made only in connection with birds ( chicken broilers or other birds). The comparison with the human patients or mice is not appropriate here.
5) The references must  be written in agreement with MDPI rules, as follows:
Journal Articles:
1. Author 1, A.B.; Author 2, C.D. Title of the article. Abbreviated Journal Name Year, Volume, page range.
Books and Book Chapters:
2. Author 1, A.; Author 2, B. Book Title, 3rd ed.; Publisher: Publisher Location, Country, Year; pp. 154–196.
3. Author 1, A.; Author 2, B. Title of the chapter. In Book Title, 2nd ed.; Editor 1, A., Editor 2, B., Eds.; Publisher: Publisher Location, Country, Year; Volume 3, pp. 154–196.
Unpublished materials intended for publication:
4. Author 1, A.B.; Author 2, C. Title of Unpublished Work (optional). Correspondence Affiliation, City, State, Country. year, status (manuscript in preparation; to be submitted).
5. Author 1, A.B.; Author 2, C. Title of Unpublished Work. Abbreviated Journal Name year, phrase indicating stage of publication (submitted; accepted; in press).
Unpublished materials not intended for publication:
6. Author 1, A.B. (Affiliation, City, State, Country); Author 2, C. (Affiliation, City, State, Country). Phase describing the material, year. (phase: Personal communication; Private communication; Unpublished work; etc.)
Conference Proceedings:
7. Author 1, A.B.; Author 2, C.D.; Author 3, E.F. Title of Presentation. In Title of the Collected Work (if available), Proceedings of the Name of the Conference, Location of Conference, Country, Date of Conference; Editor 1, Editor 2, Eds. (if available); Publisher: City, Country, Year (if available); Abstract Number (optional), Pagination (optional).
Thesis:
8. Author 1, A.B. Title of Thesis. Level of Thesis, Degree-Granting University, Location of University, Date of Completion.
Websites:
9. Title of Site. Available online: URL (accessed on Day Month Year).
Unlike published works, websites may change over time or disappear, so we encourage you create an archive of the cited website using a service such as WebCite. Archived websites should be cited using the link provided as follows:
10. Title of Site. URL (archived on Day Month Year).  

I give for this manuscript major revision, but I am sure that the authors can solve the above mentioned aspects.

Author Response

Response to reviewer 1

Dear reviewer:

Thank you very much for taking the time to review my manuscript. Please find the detailed responses below and the corresponding revisions highlighted in yellow of the re-submitted files. Regarding your question that introduction and discussion need to be related to birds, I have made concentrated revisions. However, research on the regulation of inflammatory response by gut microbiota through NLRP3 inflammasomes has not yet been found in birds. My study was the first to discover this phenomenon in chickens and validated it using cecal microbiota transplantation and antibiotic treatment. So in the discussion, we used mice to introduce this question and carry out the following discussion. Therefore, in the discussion, I kept a few studies on mice. If you still feel it's not appropriate, I will make the necessary modifications. Thank you for your valuable feedback.

Point-by-point response to comments and suggestions for authors

Comment: 1) At the introduction: the authors must perform an introduction in connection with their experiments (they must consider only the studies conducted on birds, not on human patients and not by mice). The introduction must be rewritten.

Response: I have made centralized modifications to address this issue.

Comment: 2) Before the beginning of the chapter entitled ''Materials and Methods'' the authors must write the objectives of the experimental studies presented in the manuscript.

Response: I added the objectives of the experimental studies presented in the manuscript before the beginning of the chapter entitled ''Materials and Methods''

Comment: 3) At results: Figure 4 is unclear. The authors must upload here a figure with a high resolution.

Response: I uploaded a new figure 4 with a high resolution. The resolution of original image is high. I have placed it in Figure 4 outside the manuscript, hoping that the editor can clearly integrate it into the manuscript during proofreading.  

Comment: 4) The chapter entitled ''Discussions'' must be completely rewritten. That is because, in the experimental part, the authors have performed their studies only on chicken broilers (birds), so the discussions must be made only in connection with birds (chicken broilers or other birds). The comparison with the human patients or mice is not appropriate here.

Response: I have made centralized modifications to address this issue.

Comment: The references must be written in agreement with MDPI rules

Response: I made uniform modifications to the format of all references according to MDPI rules.

Reviewer 2 Report

Comments and Suggestions for Authors

This paper found that exposing chickens to more light led to changes in the microbiome and activation of an inflammasome. Replacement of the microbiome could inhibit the inflammasome and prevent the damage, suggesting the microbiome changes are causative for the inflammation and damage and not just correlated. That's an interesting finding.

The sample size and methods seem fine, but I have a major concern about the methods as depicted in Table 1. Why is the formal experiment phase for 18L:6D using 23L:1D? Was that a mistake? Did you copy it from Heike et al. 2019?
I also am unsure of why the adaptive phases were 24L and not 12:12. Are these methods based on prior, published papers?

The discussion needs some revision at the start. A discussion should start by clearly stating what your paper found, and then tying your findings with those of past research, then ending with future directions. Restating material from the introduction is not needed. Stating findings but not explicitly clarifying whether they come from your work or a previous paper is confusing. You must very clearly and directly state that you found X, previous papers also found X or instead found Y, and what this means.

Other comments
10 I don't fully agree with the phrasing that "light pollution" can affect intestinal inflammation: the intestine is not normally exposed to light, as it's inside the body. Is it the light itself causing problems, or the lack of sleep / altered circadian rhythm? You should also account for this in the introduction paragraph 2.
16-18 The standard output of software like QIIME2 or MajorBioCloud should probably be translated into normal language for the abstract.
20 I think you need to delete the dash between light and induced
114 This is the first use of "PBS" and it must be defined. If you mean PBS solution, then explicitly state that this is a negative control. I also recommend defining CMT here, and not in the heading.
120 "we" is used, though msot of the methods are passive voice
Table 2 - Delete the hyperlinks
Figrue 3: You might as well replace P1 and P2 with "12L:12D" and "18L:6D" in the figures themselves.
164-165 Is there a standard protocol for MajorBio Cloud, or something else you need to add to the methods?
202 Replace "Family_XIII_UCG-001 norank_f__norank_o__Clostridia_vadinBB60_group," with "Clostridia vadin group" ; "Defluviitaleaceae_UCG-011" with "unclassified Defluviitaleaceae" ; and "unclassified_o__Oscillospirales" with "unclassified Oscillospirales"  [here and elsewhere in the paper]
Figure 5 A-D and 6 A-C  I cannot interpret these images. Can you highlight the signs of damage with arrows or something?
269-275 I personally do not like seeing introductory material in the discussion. The reader is unlikely to have forgotten it and does not a reminder here. Move this text to the introduction and start the Discussuon v with a summary of what your paper found.
278-290 It is not clear how any of this paragraph relates to your experiment, so perhaps move it all to the introduction.
292-297 So this is not your data, but a prior paper's? You should clearly state that with the phrase "In a previous study," and then explain how that data is similar or different from yours, and what that means.
304 "The results" yours, or Li et al's? It is not clear. Clearly state "The results of this paper" if you mean the results of this paper or "The results from these experiments" if you are tying your paper together with others.

Comments on the Quality of English Language

English editing is needed

Author Response

Response to reviewer 2

Dear reviewer:

Thank you very much for taking the time to review my manuscript. Please find the detailed responses below and the corresponding revisions highlighted in yellow of the re-submitted files.

Point-by-point response to comments and suggestions for authors

Comment: The sample size and methods seem fine, but I have a major concern about the methods as depicted in Table 1. Why is the formal experiment phase for 18L:6D using 23L:1D? Was that a mistake? Did you copy it from Heike et al. 2019? I also am unsure of why the adaptive phases were 24L and not 12:12. Are these methods based on prior, published papers?

Response: Sorry, it was my mistake. The photoperiod of the 18L:6D group in the formal experiment is 18L:6D, not 23L;1D. I have made modification in Table1. In addition, the 24Lof the adaptive phase is designed to help broiler chickens to adapt to the surrounding environment and reduce stress.

Comment: The discussion needs some revision at the start. A discussion should start by clearly stating what your paper found, and then tying your findings with those of past research, then ending with future directions. Restating material from the introduction is not needed. Stating findings but not explicitly clarifying whether they come from your work or a previous paper is confusing. You must very clearly and directly state that you found X, previous papers also found X or instead found Y, and what this means.

Response: Thank you very much for your suggestion, which has been of great help to me in writing my SCI paper. In addition, I have optimized my discussion.

Comment: 10 I don't fully agree with the phrasing that "light pollution" can affect intestinal inflammation: the intestine is not normally exposed to light, as it's inside the body. Is it the light itself causing problems, or the lack of sleep / altered circadian rhythm? You should also account for this in the introduction paragraph 2.

Response: I supplemented an explanation of sleep deprivation and circadian rhythm disorder in the paragraph 2 as following: Recently, an increasing number of studies have shown that prolonged exposure light leads to intestinal dysfunction. It was found that extended exposure to light (24L) disrupted gut barrier integrity through sleep deprivation and circadian and circadian rhythm disorder, which induced the inflammation of colon in mice (Eum et al., 2023; Polidarová et al., 2017). Prolonged exposure to light can disturbed circadian rhythm and rapidly proliferate and differentiate Th17 cells in the intestine, and secrete large amounts of inflammatory cytokines, which leading to intestinal inflammation in mice (Yu et al., 2013).

Comment: 16-18 The standard output of software like QIIME2 or MajorBioCloud should probably be translated into normal language for the abstract.

Response: The MajorBio Cloud platform is data analysis platform, where we analyze raw data.

Comment: 20 I think you need to delete the dash between light and induced

Response: I have deleted all dashes between light and induced throughout the manuscript.

Comment: 114 This is the first use of "PBS" and it must be defined. If you mean PBS solution, then explicitly state that this is a negative control. I also recommend defining CMT here, and not in the heading.

Response: The 18L:6D+PBS group was negative control group, and I explained this on Line 113. In addition, I defined the CMT on Line 114.

Comment: 120 "we" is used, though most of the methods are passive voice

Response: I changed the “In the 18L:6D+CMT group, we transplanted the gut microbiota of broiler chickens from 12L:12D group to the broiler chickens of 18L:6D.” to “In the 18L:6D+CMT group, the gut microbiota of broiler chickens from 12L:12D group were transplanted to the broiler chickens of 18L:6D.”.

Comment: Table 2 - Delete the hyperlinks

Response: I deleted the hyperlinks in the Table 2.

Comment: Figure 3: You might as well replace P1 and P2 with "12L:12D" and "18L:6D" in the figures themselves.

Response: Sorry, the format of “12L:12D and 18L:6D” cannot be used on the analysis platform, so I cannot modify them. In addition, I indicated that P1 referred to 12L:12D group and P2 referred to 18L:6D group in the figure caption.

Comment: 164-165 Is there a standard protocol for MajorBio Cloud, or something else you need to add to the methods?

Response: The MajorBio Cloud platform is data analysis platform, where we analyze raw data without a standard protocol.

Comment: 202 Replace "Family_XIII_UCG-001 norank_f__norank_o__Clostridia_vadinBB60_group," with "Clostridia vadin group" ; "Defluviitaleaceae_UCG-011" with "unclassified Defluviitaleaceae" ; and "unclassified_o__Oscillospirales" with "unclassified Oscillospirales"  [here and elsewhere in the paper]

Response: I replaced "Family_XIII_UCG-001 norank_f__norank_o__Clostridia_vadinBB60_group" with "Clostridia vadin group"; "Defluviitaleaceae_UCG-011" with "unclassified Defluviitaleaceae"; and "unclassified_o__Oscillospirales" with "unclassified Oscillospirales" throughout the manuscript.

Comment: Figure 5 A-D and 6 A-C I cannot interpret these images. Can you highlight the signs of damage with arrows or something?

Response: In the figure 5A-D and 6A-C, I highlighted the signs of damage with arrows and explained the damage indicated by the arrows in the figure caption.

Comment: 269-275 I personally do not like seeing introductory material in the discussion. The reader is unlikely to have forgotten it and does not a reminder here. Move this text to the introduction and start the Discussion with a summary of what your paper found.

Response: I moved the content of P269-275 of the discussion to the introduction and start the discussion with a summary of what my paper found.

Comment: 278-290 It is not clear how any of this paragraph relates to your experiment, so perhaps move it all to the introduction.

Response: I moved this paragraph to the introduction.

Comment: 292-297 So this is not your data, but a prior paper's? You should clearly state that with the phrase "In a previous study," and then explain how that data is similar or different from yours, and what that means.

Response: The content of P292-297 is the result of this study, and I explained it in this sentence.

Comment: 304 "The results" yours, or Li et al's? It is not clear. Clearly state "The results of this paper" if you mean the results of this paper or "The results from these experiments" if you are tying your paper together with others.

Response: “The results” is mine. I changed “The results” to “The results of this study”.

Round 2

Reviewer 1 Report

Comments and Suggestions for Authors

Generally, the manuscript is more concise; now the reader understands that the introduction and discussion section are connected with birds(chicken broiler).  Two minor corrections are needed before publication:

1) Figure 4 is not clear. Authors must provide a picture with a high quality here.

2) References appear not to be written according to MDPI rules. The references must be written in the following way: 

  • Journal Articles:
    1. Author 1, A.B.; Author 2, C.D. Title of the article. Abbreviated Journal Name YearVolume, page range.
  • Books and Book Chapters:
    2. Author 1, A.; Author 2, B. Book Title, 3rd ed.; Publisher: Publisher Location, Country, Year; pp. 154–196.
    3. Author 1, A.; Author 2, B. Title of the chapter. In Book Title, 2nd ed.; Editor 1, A., Editor 2, B., Eds.; Publisher: Publisher Location, Country, Year; Volume 3, pp. 154–196.
  • Conference Proceedings:
    7. Author 1, A.B.; Author 2, C.D.; Author 3, E.F. Title of Presentation. In Title of the Collected Work (if available), Proceedings of the Name of the Conference, Location of Conference, Country, Date of Conference; Editor 1, Editor 2, Eds. (if available); Publisher: City, Country, Year (if available); Abstract Number (optional), Pagination (optional).
  • Thesis:
    8. Author 1, A.B. Title of Thesis. Level of Thesis, Degree-Granting University, Location of University, Date of Completion.

Author Response

Response to reviewer 1

Dear reviewer:

Thank you very much for taking the time to review my manuscript.

For the Figure 4, I uploaded a new figure 4 with a high resolution. The resolution of original image is high. I have placed it in Figure 4 outside the manuscript, hoping that the editor can clearly integrate it into the manuscript during proofreading. The picture is quite compact, so it needs to be enlarged to see it

For the reference, I made uniform modifications to the format of all references according to MDPI rules.